# Effects of Nutrition on Cognitive Function in Adults with or without Cognitive Impairment: A Systematic Review of Randomized Controlled Clinical Trials

**DOI:** 10.3390/nu13113728

**Published:** 2021-10-22

**Authors:** Laia Gutierrez, Alexandre Folch, Melina Rojas, José Luis Cantero, Mercedes Atienza, Jaume Folch, Antoni Camins, Agustín Ruiz, Christopher Papandreou, Mònica Bulló

**Affiliations:** 1Nutrition and Metabolic Disorders Research Group, Department of Biochemistry and Biotechnology, Rovira i Virgili University, 43201 Reus, Spain; laia.gutierrez@iispv.cat (L.G.); alexandre.folch@urv.cat (A.F.); melinaisabella.rojas@urv.cat (M.R.); jaume.folch@urv.cat (J.F.); 2Nutrition and Metabolic Disorders Research Group, Institute of Health Pere Virgili—IISPV, 43204 Reus, Spain; christoforos.papandreou@iispv.cat; 3Laboratory of Functional Neuroscience, Pablo de Olavide University, 41013 Seville, Spain; jlcanlor@upo.es (J.L.C.); matirui@upo.es (M.A.); 4CIBERNED, Network Center for Biomedical Research in Neurodegenerative Diseases, 28031 Madrid, Spain; camins@ub.edu (A.C.); aruiz@fundacioace.com (A.R.); 5Department of Pharmacology, Toxicology & Therapeutic Chemistry, Faculty of Pharmacy & Food Sciences, University of Barcelona, 08028 Barcelona, Spain; 6Institut de Neurociències (UBNeuro), University of Barcelona, 08035 Barcelona, Spain; 7ACE Alzheimer Center Barcelona, Universitat Internacional de Catalunya (UIC), 08028 Barcelona, Spain; 8CIBER Physiology of Obesity and Nutrition (CIBEROBN), Carlos III Health Institute, 28029 Madrid, Spain

**Keywords:** diet, food, supplements, dietary interventions, cognitive impairment, healthy, subjective cognitive decline, Alzheimer’s disease

## Abstract

New dietary approaches for the prevention of cognitive impairment are being investigated. However, evidence from dietary interventions is mainly from food and nutrient supplement interventions, with inconsistent results and high heterogeneity between trials. We conducted a comprehensive systematic search of randomized controlled trials (RCTs) published in MEDLINE-PubMed, from January 2018 to July 2021, investigating the impact of dietary counseling, as well as food-based and dietary supplement interventions on cognitive function in adults with or without cognitive impairment. Based on the search strategy, 197 eligible publications were used for data abstraction. Finally, 61 articles were included in the analysis. There was reasonable evidence that dietary patterns, as well as food and dietary supplements improved cognitive domains or measures of brain integrity. The Mediterranean diet showed promising results, whereas the role of the DASH diet was not clear. Healthy food consumption improved cognitive function, although the quality of these studies was relatively low. The role of dietary supplements was mixed, with strong evidence of the benefits of polyphenols and combinations of nutrients, but with low evidence for PUFAs, vitamin D, specific protein, amino acids, and other types of supplements. Further well-designed RCTs are needed to guide the development of dietary approaches for the prevention of cognitive impairment.

## 1. Introduction

Dementia represents a serious public health challenge for elderly people [1], while Alzheimer’s disease (AD), the most common type of dementia, has become one of the biggest mental burdens [2]. Except for the recent and controversial approval of Aduhelm to treat patients with Alzheimer’s disease, there are no effective treatments to prevent or delay its progression, only for treating the symptoms of mild to moderate dementia [3]. Therefore, there is an urgent need to identify effective strategies to prevent dementia. It is well recognized that mild cognitive impairment (MCI) precedes AD [4,5]; however, new evidence suggests that subtle and silent pathological brain changes associated with subjective decline are also present in subjects with subjective cognitive decline (SCD), defined as a self-reported decline in cognitive performance compared to an individual’s previous level of functioning, which cannot be determined by neuropsychological tests [6] and precedes objective cognitive decline [7,8,9].

Lifestyle choices based on a healthy dietary management strategy, either with dietary patterns, food, and dietary supplements may be effective in preventing cognitive impairment. Since inflammation and oxidative stress have long been considered to play a major role in cognitive impairment and AD [10], most of the dietary intervention trials conducted so far involved foods or dietary supplements with antioxidant and anti-inflammatory properties. Regarding dietary patterns, there is a growing body of evidence from epidemiological studies [11,12] and randomized controlled trials (RCTs) [13] that suggest a neuroprotective role by the Mediterranean diet, the Dietary Approaches to Stop Hypertension (DASH) diet, and the Mediterranean-DASH diet Intervention for Neurodegenerative Delay (MIND) diet in reducing cognitive decline [14]. Furthermore, evidence from observational [15] and interventional studies [16,17] suggest that certain food groups included in these dietary patterns, such as fruits and vegetables, legumes, whole grains, nuts, and olive oil, may improve cognitive functioning. Specific nutrients like unsaturated fatty acids and antioxidants have also been associated with a decreased risk of cognitive decline [18]. In this sense, a 10-year prospective study conducted over 1640 older adults free from dementia at baseline found that dietary flavonoid intake was associated with better cognitive functioning at baseline and a lower risk of cognitive decline in the follow-up period [19]. Although low vitamin D has also been related to worse cognitive performance and cognitive decline, there is no evidence that vitamin D supplementation has any real benefit for cognitive functioning yet [20].

Despite the growing interest in the ability of different dietary approaches to favorably affect cognition, existing evidence from RCTs investigating the effectiveness of healthy diets, specific healthy foods, or dietary supplements on cognitive functioning has yielded inconsistent results with a high heterogeneity between trials. A systematic review of RCTs published between 2016 and 2018 linking diet to cognitive functioning in MCI patients found serious inconsistencies regarding the potential neuroprotective effect of different vitamins, mainly due to the heterogeneity of dietary interventions and measures used to assess cognitive health [21]. Similarly, another systematic review of RCTs published between 2014 and 2017, conducted in cognitively healthy older adults, found a moderate evidence of beneficial effects of dietary interventions on specific cognitive domains [22]. Although both, cognitively normal older adults and MCI patients benefited primarily from B-vitamin supplementation, the former also showed some benefit when they were supplemented with long-chain polyunsaturated fatty acids (PUFAs) [21,22]. Finally, the systematic review of the literature up to 2017, based on the cognitive effects following different dietary patterns in adults over 60 years, revealed that the Mediterranean diet is the most investigated pattern with the potential to protect against cognitive impairment, whereas the evidence provided for MIND, DASH, anti-inflammatory and prudent healthy, although more limited, also showed promising results on cognitive outcomes [13].

Hence, to comprehensively update the evidence, we undertook a systematic review of the newest RCTs evaluating the impact of dietary counselling interventions, food-based interventions and dietary supplementations on cognitive function in younger and older adults with or without cognitive impairment. This work has the potential to establish updated evidence-based research for potential nutritional strategies for the management of cognitive impairment and contributes to define the scope for further nutritional research in the field of cognition.

## 2. Materials and Methods

This systematic review was conducted following the Preferred Reporting Items for Systematic reviews and Meta-Analyses (PRISMA) guidelines (http://www.prisma-statement.org/ (accessed on 21 October 2021). Refs. [23,24] and the review protocol was registered with PROSPERO (PROSPERO 2021: CRD42021245941).

### 2.1. Eligibility Criteria

The eligible studies had to fulfill the following criteria: (1) they were RCTs, including crossover and parallel designs; (2) they assessed the impact of a dietary intervention (i.e., diet counseling intervention, food-based intervention, or dietary supplement trial) in adults with or without cognitive impairment but without previous diagnoses of AD; (3) they were published in English and available in PubMed between 1 January 2018 and 31 July 2021. The reason for the selection of this period was that previous systematic reviews had been performed on this topic before [13,21,22]. Studies were excluded if they involved multifactorial lifestyle interventions, in which effects of dietary factors could not be distinguished from other lifestyle factors (such as multidomain or physical training intervention); they used probiotics in the form of pills (not in a food matrix); or they were observational studies, case reports, comments, letters, editorials, in vitro studies, animal studies, duplicate studies, narrative or systematic reviews and study protocols. Additionally, assessment of cognitive functioning could include the application of different neuropsychological batteries and/or specific questionnaires, as well as the assessment of brain integrity, as revealed by structural, functional and metabolic changes. Studies containing combined interventions (i.e., nutritional plus physical activity or behavioral interventions) were excluded unless the nutritional effect on the outcomes was analyzed separately from other interventions. When there was more than one study of the same trial, the publication with the largest sample size and longest follow-up was selected.

### 2.2. Literature Search

A literature search was carried out using the MEDLINE-PubMed database. The search was limited to the period of 1 January 2018 to 31 July 2021. For this we employed the algorithm described in Supplementary Files. The search strategy used keywords and selected terms related to diet, specific food or food groups, nutrients and nutritional supplements, combined with terms related to cognitive impairment, Alzheimer’s disease, and dementia.

### 2.3. Screening and Data Extraction

Screening of eligible studies and data extraction were performed independently by three independent reviewers (L.G., A.F., and M.R.), and disagreement between the three authors was resolved by consulting senior researchers (C.P. and M.B.).

Searching for duplicate publications was performed using an electronic database and then the full text of each potentially relevant study was reviewed. A specific grid was adopted to facilitate data extraction and selection, including the following information: the author’s surname, year of publication, country where the study was conducted, characteristics of the trial participants (cognitive health, sample size based on participants randomized, and age), study design (including the design of a parallel or crossover intervention and blinding), duration, type of the intervention and the comparator (including dose of exposure), study outcomes and major findings. Cognitive function was assessed by using different neuropsychological batteries and/or specific questionnaires, as well as measures of brain integrity, as revealed by structural, functional and metabolic changes. Neuropsychological batteries and questionnaires included Montreal Cognitive Assessment (MoCA), Mini-Mental State Exam (MMSE), Trail Making Test (TMT), Clock Drawing Test (CDT), Wechsler Adult Intelligence Scale (WAIS), Wechsler Memory Scale (WMS), Rey Auditory Verbal Learning Test (AVLT), Dementia Rating Scale (DRS), Wisconsin Card Sorting Test (WCST), Alzheimer’s Disease Assessment Scale-Cognitive Subscale test (ADAS-Cog), the Cambridge Neuropsychological Test Automated Battery (CANTAB), the Consortium to Establish a Registry for Alzheimer’s Disease neuropsychological battery (CERAD.NB), etc. Regarding brain integrity, fMRI methods assessing blood oxygen level-dependent (BOLD) and (18F) FDG PET scans were considered to assess brain glucose metabolism.

### 2.4. Quality Assessment

The quality assessment was based on extracted data using the second version of the Cochrane Risk-of-Bias instrument (RoB 2) [25] and discrepancies were resolved by consensus between two additional authors. Risk-of-Bias was assessed in terms of the following domains: (1) randomization process; (2) deviation from intended interventions; (3) missing outcomes data; (4) measurement of the outcome; (5) selective outcome reporting; and (6) overall bias. Each domain was considered to be of “low risk”, “some concern”, or “high risk” of bias. According to RoB 2, a result of “high risk” for any individual domain leads to an overall high risk of bias, while a result of “some concern” in any individual domain is judged as an overall risk of “some concern”. If multiple domains are problematic, confidence in the outcome decreases and the overall risk should be considered “high” [25,26].

### 2.5. Data Synthesis

Studies were grouped according to their design (i.e., food-based intervention, dietary counselling intervention or dietary supplement intervention) and secondly according to outcome. The review emphasized the impact of the identified nutritional interventions on participants’ cognitive health, which was assessed through changes in mean values of measures of cognitive function, standard deviation or 95% confidence interval, odds ratio and/or relative risk.

## 3. Results

### 3.1. Search and Selection of Studies

Through a database search, 197 articles were identified (Figure 1). After examining titles, abstracts and full-texts, 83 articles were considered to be out of the scope of this review and were excluded. Further screening was performed by reading full text and 53 articles were non-eligible due to: participants having a diagnosis of AD (n = 2), studies without nutritional interventions (n = 12), multi-domain interventions (n = 14), not assessing cognition (n = 16), interventions administrating drugs or probiotics in pill form (n = 6), and manuscripts with commentaries or protocols (n = 3). Finally, a total of 61 articles met the inclusion criteria and were finally considered in the present systematic review.

### 3.2. Characteristics of Included Studies

Characteristics of the articles included in the systematic review are shown in Table 1. There were 5 dietary counseling interventions, 7 food-based interventions and 49 dietary supplement interventions. With respect to the study design, 52 articles had a parallel design and 9 had a crossover design. The duration of the interventions ranged from 1 day to 4 years and the age of the participants ranged from 18 to 90 years old. Trials were conducted in 18 countries including Australia (n = 6), Belgium (n = 1), China (n = 3), Cyprus (n = 1), Finland (n = 1), France (n = 4), Germany (n = 3), Greece (n = 1), Japan (n = 17), Malaysia (n = 2), Netherlands (n = 1), Portugal (n = 1), South Korea (n = 3), Spain (n = 1), Sweden (n = 1), UK (n = 2), and USA (n = 11), and two multicenter studies in Spain–USA (n = 1) and France–Canada (n = 1).

### 3.3. Diet Counselling Interventions (n = 5)

The effect of the Mediterranean diet enriched with a single food was evaluated in 3 different studies. A 8-week RCT with a crossover design revealed improvements in processing speed in 33 older cognitively healthy subjects who followed a Mediterranean diet enriched with fresh lean pork (2–3 servings/week) compared to a low-fat diet [27]. Similarly, the same researchers evaluated the effect of a Mediterranean diet enriched with 3–4 servings/day of dairy products in 43 older cognitively healthy subjects, showing a significant enhancement in processing speed in the Mediterranean diet group compared to a low-fat diet group [28]. The effect of a Mediterranean diet enriched with either high phenolic extra virgin olive oil (50 mL/d) or moderate phenolic extra virgin olive oil (50 mL/d), or a Mediterranean diet alone for 12 months was also evaluated in 50 Greek older adults during 12 months. Participants allocated in both olive oil intervention groups showed significant improvements in cognitive function compared to the control group, with a greater effect in the high phenolic extra virgin olive oil consumption group [29]. Additionally, a 6-month RCT was conducted in 79 adults with cognitive impairment but no dementia and those who followed the DASH diet did not reveal significant improvements on general cognition [30]. The influence of adhering to Finnish Nutrition Recommendations for 4 years was also tested in men and women aged 57–78 in a parallel RCT with no beneficial effects on cognition compared to the control group [31].

### 3.4. Food-Based Interventions (n = 7)

Six food-based intervention studies were conducted on cognitively healthy participants. In a parallel intervention study carried out in 61 subjects over 60 years, the consumption of 3 g/d of matcha green tea powder for 12 weeks improved cognitive and memory function [32]. Similarly, daily avocado consumption (140–175 g/d) for 3 months ameliorated attentional inhibition in 47 young adults [33]. In a longer trial conducted over 52 older adults, consumption of 100 g/d of ultra-high hydrostatic pressurized brown rice for 2 years reduced overall cognitive decline compared to white rice intake [34]. Similarly, a crossover study of 31 older adults showed a reduction in overall cognitive decline after consuming 3 servings/d of dewaxed brown rice for 6 months [35]. In contrast, an acute crossover study comparing the consumption of rye-based bread with white wheat flour bread for 3 days did not find any effect neither in working memory nor in attention [36]. In a recent long-term (2 years) trial with 708 older participants, the consumption of 60 g/d walnuts had no any beneficial effect in global cognition [37]. Only one study has evaluated the effect on animal-based food consumption on cognition. In this regard, the consumption of 33.4 g/d of camembert cheese or the same amount of processed cheese made from mozzarella and cream cheese for 3 months did not produce improvements in general cognitive functions in 65 subjects with MCI [38].

### 3.5. Dietary Supplement Interventions (n = 49)

Forty-nine trials were categorized according to the type of supplementation into proteins and amino acids, vitamins, polyphenols, PUFAs, combinations of nutrients and other supplements.

#### 3.5.1. Protein and Amino Acid Supplements (n = 7)

A randomized, double-blind, placebo-controlled 12-week trial evaluated the effect of combined anserine and carnosine supplementation (750:250 mg/d) in 25 MCI subjects. Those subjects assigned to the active group showed a better response to the global Clinical Dementia Rating (gloCDR), without showing any beneficial effect on the other psychometric tests including the Mini-Mental State Examination (MMSE), the Wechsler Memory Scale, and the Alzheimer’s Disease Assessment Scale (ADAS). However, when APOE4+ or APOE4- subjects were separately analyzed, improvements in MMSE and gloCDR were observed in the APOE4+ subjects [39]. A study conducted over 268 community-dwelling adults without dementia showed that administration of 200 μg/d of a casein-derived peptide Met-Lys-Pro (MKP) for 24 weeks significantly improved the ADAS-Cog subscale, but not the ADAS-Cog total score compared to the placebo group [40]. Similarly, the intake of lactotripeptide tablets containing 1.4 mg/d valine-proline-proline and 2.0 mg/d isoleucine-proline-proline for 8 weeks also did not produce significant improvements in cognitive assessments in 35 middle-aged and older adults [41]. A parallel trial investigating the effect of taurine supplementation (150 mL/d water containing 1.5 g taurine) for 14 weeks also failed to show significant benefits on cognition in older women [42]. The only crossover trial with amino acid supplementation conducted in 20 young volunteers receiving either 125 mg/d of L-homoarginine for 4 weeks or placebo, did not find changes neither in vascular nor in neuronal function in young volunteers [43]. The effect of 1 g/d of whey peptide supplementation for 12 weeks resulted in higher verbal fluency test (VFT) scores in the intervention group compared with the placebo. A further subgroup analyses according to the degree of subjective fatigue showed that changes in the VFT as well as the Stroop and subjective memory function tests were significantly better in subjects with high-level fatigue from the whey peptide group [44]. Finally, another RCT was conducted on 19 male subjects consuming either β-alanine (12 g/d) or placebo for 14 days prior to a 24-h simulated military training (SUSOP) during which participants received cognitive assessment. Individuals in the placebo group showed worse performance in the visuomotor training task compared to the active group [45].

#### 3.5.2. Vitamin Supplements (n = 4)

Three RCTs evaluated the effect of vitamin D on cognition and one study evaluated B group vitamins. A 3-year trial conducted in 260 older postmenopausal African-American women showed no significant difference in cognition over time, based on the MMSE score, after the intake of D3 vitamin (daily dose of 2400 IU, 3600 IU, or 4800 IU to achieve and maintain serum levels of 30 ng/mL) [46]. A study in 181 subjects with MCI evaluated the supplementation of a daily dose of 400 IU vitamin D3 for 12 months and demonstrated an overall significant improvement of cognitive functioning in the vitamin D3 group compared to placebo [47]. Another 6-month double-blind, four-arm placebo, controlled trial evaluated the effects of vitamin D supplementation delivered either as enhanced vitamin D2 (UV-exposed mushroom, D2M) or D3 (synthetic) for 24 weeks compared to standard mushroom (SM) and placebo (PL) controls on cognition and mood in 436 healthy subjects aged over 60 years. There were no significant effects of treatment on any of the measures of cognitive function or mood, thus not supporting any benefit of vitamin D supplementation for mood or cognition [48].

Regarding B-group vitamins, vitamin B9 (400 µg/d) and vitamin B12 (25 µg/d) were administered alone or in combination (400 µg/d + 25 µg/d B9 and B12, respectively) to 180 subjects with MCI for 6 months. The folic acid plus vitamin B12 supplementation significantly improved the Full-Scale IQ, Information and Digit Span scores. Although the supplementation with vitamin B9 and vitamin B12 significantly improved cognition, the combination of folic acid and vitamin B12 was superior to either folic acid or vitamin B12 alone [49].

#### 3.5.3. Polyphenols Supplements (n = 10)

The acute consumption of cocoa flavanols was tested in 11 young adults with type-1 diabetes (T1D) and their matched healthy controls (15 mg in healthy and 900 mg in T1D subjects) 2 h before a flanker test. The authors found faster reaction times and greater BOLD signal amplitude in the supramarginal gyrus and inferior frontal gyrus compared to the placebo in both groups [50]. In another larger trial including 101 healthy adults aged 40–60 years with overweight and obesity, the long-term (24 weeks) supplementation with anthocyanin-rich Aronia melanocarpa extract (90 mg and 150 mg) shown a significant improvement in psychomotor speed but not in the rest of cognitive tests [51].

One of the target populations for polyphenol supplementation was healthy older adults. In this case, curcumin supplementation (80 mg/d curcumin) in 38 subjects showed a beneficial effect on working memory after 12 weeks, as well as improved mood and lower fatigue at both 4 and 12 weeks of the trial [52]. Similarly, the consumption of 258 mg/d of flavonoids from a polyphenol-rich extract of grape and blueberry for 6 months in 92 subjects improved age-related episodic memory impairment in those subjects with greater cognitive impairment [53]. The effect of consuming two cups of Montmorency tart cherry juice improved memory and performance on learning tasks after a 12-week randomized controlled trial in 37 older adults [54].

One double-blind controlled trial was performed in 90 young subjects with SCD. The effect of the spearmint extract intake was evaluated at two different doses (600 or 900 mg/d) for 90 days. Results indicated that subjects consuming 900 mg/d dose significantly improved working memory compared to the placebo group [55].

There were also four studies with 12-week interventions, one with Lactobacillus plantarum C29-fermented soybean consumption (800 mg/d) showing enhanced cognitive performance and attention in a parallel trial in 100 MCI subjects [56], another with Cosmos caudaus supplement (500 mg/d) that potentially improved global cognition function in 23 MCI patients [57]. A crossover trial evaluated the effect of continuous intake of chlorogenic acid (1.107 g/d) in 28 MCI patients, showing an enhanced attention and executive function [58]. Additionally, a 6-month trial conducted in 36 MCI patients investigated the effect of Biokesum, a registered Persicaria minor extract supplement rich in polyphenols (500 mg/d), supplementation and observed improved visual memory, negative mood and bilateral dorsolateral prefrontal cortex activation [59].

#### 3.5.4. PUFAs Supplements (n = 5)

A double-blinded placebo-controlled parallel trial enrolled 555 officers who daily consumed 8 capsules of krill oil (2.3 g/d of omega-3) or macadamia nut oil containing between 53% and 67% oleic acid and 16% and 24% palmitoleic acid (control group) for 20 weeks. The results did not support a significant beneficial effect of supplemented krill-based omega-3 fatty acids on cognitive function [60]. In the MAPT study, 1680 French community-dwellers aged 70 or over reporting subjective memory complaints received either w-3 PUFAs supplementation (800 mg/d of DHA and ≤225 mg/d of EPA) alone, a multidomain intervention (nutrition, physical activity and cognitive training), w-3 PUFAs supplementation plus the multidomain intervention or placebo for 3 years. Participants who consumed PUFAs had no significant improvement in cognition compared to placebo [61,62,63]. Moreover, 33 of the participants from the MAPT study underwent 2-(18F)fluoro-2-deoxy-D-glucose (FDG) PET and received either V0137 (800 mg/d of an active supplement containing a minimum of 65% DHA and a maximum of 15% EPA) or placebo for 12 months, although non-significant effects on cognition were observed [64].

#### 3.5.5. Combination of PUFAs, Polyphenols and Vitamins (n = 8)

The effect of combining PUFAs, polyphenols and vitamins has also been evaluated. The consumption of omega-3 PUFAs, catechins and ginsenosides (960 mg/d EPA, 624 mg/d DHA, 16 mg/d ginsenosides and 26 mg/d green tea catechins) for 1 month induced enhanced cognitive functioning and increased fMRI functional connectivity during task execution in 10 healthy older adults [65]. In contrast, combining omega-3 PUFAs with vitamin E (1.72 g/d DHA and 0.6 g/d EPA, mixed with tocopherols at 11.2–18 mg/g) exerted a small negative effect on psychomotor speed but a significant increased perception of self-reported cognitive failures compared to participants in the control group who were consuming 4 capsules/d, each containing 990 mg of low polyphenol olive oil and 10 mg of fish oil, equivalent to 1.8 mg EPA and 1.2 mg DHA [66]. Eighteen older adults with MCI supplemented with a combination of omega-3 and omega-6 fatty acids, and vitamins (4.14 g DHA/0.810 mg EPA, 1.8 g GLA/3.15 g LA, 0.6 mg vitamin A, 22 mg alfa-tocopherol, 760 mg gamma-tocopherol per day) for 24 weeks, significantly reduced cognitive function and functional capacity [67]. Another trial with a three-arm design assigned 152 cognitively healthy participants to take fish oil (2 g/d DHA + 0.4 g/d EPA), or curcumin (160 mg/d), or a combination of both, or placebo for 16 weeks. The research showed that only in males was fish oil supplementation able to improve processing speed, while curcumin intake showed benefits in working memory. However, the combination of fish oil with curcumin did not produce additional benefits [68]. A similar 24-week RCT, with an additional 24-week post-intervention follow-up was carried out in 94 older adults with SCD who were assigned to consume fish oil (1.6 g/d EPA, 0.8 g/d DHA) + placebo powder (24 g/d), blueberry powder (25 g/d) + placebo oil, fish oil + blueberry powder or placebo (placebo oil + placebo powder). The study demonstrated that both treatments were associated with an enhancement of perceived functional capability that was maintained beyond the 24-week discontinuation phase as well as reduced interference in memory for those receiving blueberry powder supplementation. However, there were no positive outcomes in cognition after following the combined treatment [69]. A four-arm double-blind, placebo-controlled trial conducted in 240 MCI adults evaluated the effect of folic acid (FA) + DHA (FA 800 μg/d + DHA 800 mg/d), FA (800 μg/d), DHA (800 mg/d) and placebo for 6 months, showing that FA, DHA and their combination significantly improved cognitive function, although FA + DHA was more beneficial than each individual nutrient on their own [70]. In contrast, there were no significant differences in cognition measures among 33 cognitively healthy older adults who were provided with a vitamin B complex (1 mg of vitamin B12, 100 mg of vitamin B6 and 800 µg of folic acid per day) and randomized to either 2152 mg/d of DHA or placebo over 6 months [71]. Finally, cerebral perfusion was assessed in a subset of 13 subjects from 49 MCI patients, aged 50–80 years, who were given a combination of PUFAs and vitamin E (daily intake of 1320 mg EPA, 880 mg DHA and 15 mg vitamin E) or placebo (sunflower oil) for 26 weeks, showing an improvement of cerebral perfusion in the treatment group [72].

#### 3.5.6. Other Types of Dietary Supplements (n = 15)

Two double-blind RCTs used terpenoids as a dietary supplement with recognized antioxidant properties. Consumption of astaxanthin and sesamin (6 mg/d and 10 mg/d, respectively) for 3 months significantly improved psychomotor and processing speed in a study of 21 MCI patients [73]. In contrast, auraptene-enriched juice (6 mg/d of auraptene) for 24 weeks did not provide any extra benefit for global cognitive functioning as assessed by MMSE in 41 cognitively healthy adult subjects [74]. Other single dietary supplements or their combinations have been tested either in cognitively healthy adult subjects or in subjects with some degree of cognitive decline with contradictory results. In cognitively healthy subjects, 5 intervention studies have evaluated the effect of fungi and algae extracts, other plant extracts, or combination of minerals and vitamins. A 12-week parallel intervention study in 31 cognitively healthy adults allocated to consume either a combination of an extract of Eleutherococcussenticosus (ES) and rhizomes of Drynariafortunei (DR) (203.01 mg/d of ES leaf extract, and 20.01 mg/d of DR), showed significant improvements in language, semantic fluency and figure recall scores [75]. Consumption of Hericiumerinaceus supplement (3.2 g/d) for 12 weeks also improved cognitive function and prevented cognitive decline in a study of 34 cognitively healthy subjects over the age of 50 years [76]. The effect of fermented Laminaria japonica was also evaluated in 60 moderately active senior subjects randomized to consume either 1.5 g/d of this supplement or a sucrose pill as placebo. After 6 weeks of intervention, several neuropsychological tests were performed and a significant improvement in most of them was observed for the intervention group [77]. In contrast, another study in 44 cognitively healthy subjects (minimum age was 40 years old) who consumed 10 mL/d of an ethanol extract of traditional herbal medicines sage, rosemary and melissa) or 10 mL/d of placebo for 2 weeks, and did not find significant differences between treatment and placebo, only significant improvements to delay word recall in a subgroup analysis in subjects under 63 years [78]. Finally, an ancillary study nested within the Personalized Prevention of Colorectal Cancer Trial was conducted in 250 subjects with a calcium intake ≥700 mg/d and lower than 2000 mg/d and a Ca:Mg ratio ≥ 2.6. Participants were randomly allocated either to consume a personalized dose of Mg supplementation to reduce the Ca:Mg intake ratio around 2.3 or to the placebo group for 12 weeks. Subjects who reduced the Ca:Mg ratio by the personalized treatment significantly improved global cognitive function assessed by the MoCa test [79]. Additionally, two other RCTs were conducted in cognitively normal participants. A 6-week parallel RCT in 54 high-perfectionist men showed the consumption of milk protein concentrate drink enriched with phospholipids, sphingomyelin, phosphatidylcholine and phosphatidylethanolamine (250 mL/d with 2.7 g of phospholipids) enhanced performance on switch attention task [80]; while 33 elderly nursing home residents allocated to receive medium-chain triglycerides (6 g/d) either alone or in combination with L-Leucine (1.2 g/d) and cholecalciferol (20 μg/d) for 3 months significantly improved their cognition compared to the control group (6 g/d long-chain triglycerides) [81]. Other studies with different dietary supplements or their combinations have also evaluated their potential beneficial role on cognition in subjects with subjective cognitive decline. In this case, two 12-week studies conducted by the same investigators on subjects with subjective cognitive decline revealed improvements in verbal fluency, executive functions, memory retrieval and enhanced attention after matured hop bitter acid consumption. One of the trials was conducted in 60 adults (aged 45–64 years) who received either orally administered matured hop bitter acids (35 mg/d) derived from Humulus lupulus L, a traditional herb used in beer brewing, or placebo for 12 weeks [82]. Treatment improved verbal memory and executive function compared to participants in the placebo group [82]. The same supplement was tested in 49 adults (45–69 years), increasing the Symbol Digit Modalities Test score and also memory retrieval in the subgroup with SCD [83]. The effect of low (600 mg/d, n = 30) and high-doses (1200 mg/d, n = 30) of Tremella fuciformis (TF) consumption on cognition was evaluated in 60 young adults after 4 and 8 weeks of intervention, and subjects in the intervention group showed fewer self-reported cognitive complaints, better short-memory and executive functions compared to placebo in a non-dependent dose manner. Along with this, TF supplementation was related to increases in gray matter volumes of specific brain regions [84]. Another study conducted in 30 older subjects (60–80 y) evaluated the administration of a daily dose of 750 mg of spermidine-rich plant extract containing 1.2 mg of the natural polyamine plus 510 mg cellulose compared to the placebo containing 750 mg potato starch and 510 mg cellulose and revealed a moderate enhancement of memory performance and mnemonic discrimination ability after 3 months of intervention [85]. The combination of different dietary supplements on cognition in subjects with cognitive impairment has also been evaluated in 2 different studies with different intervention durations (6 and 12 months). The first study was a multicenter trial conducted in 99 participants randomized to receive multinutrient capsules (3 capsules/d, each capsule containing 250 mg DHA, 40 mg EPA, 5 mg vitamin E, 15 mg phosphatidylserine, 90 mg tryptophan, 125 ug vitamin B, 250 ug folate and 60 mg ginkgo biloba) or placebo for 12 months. Subjects in the active group had no significant improvement in global cognitive function when compared to the placebo group [86]. The second study was a multicenter, randomized, double-blind and placebo-controlled trial evaluating the effect of Twendee X, a patented supplement with 8 antioxidants, on MMSE and Hasegawa Dementia Scale-revised test (HDS-R) in 78 MCI patients who were randomized to the treatment or placebo group for 6 months. Cognitive assessment exhibited a significant improvement in the treatment group compared to placebo [87].

**Table 1 nutrients-13-03728-t001:** Characteristics of the 61 studies included in the systematic review.

Author, Year	Design	Country	Study Population	n	Age, y (Mean ± sd or Range)	Intervention	Control	Duration	Outcomes ^1^	Main Results
Dietary counselling interventions										
Blumenthal et al., 2019 [30]	Double-blind, parallel	USA	SCD	79	65.4 ± 6.8	DASH diet	Usual dietary and exercise habits.	6 m	TMT, the Stroop Test, the Digit Span Forward and Backward subtest and the Digit Symbol Substitution Test from the WAIS, the Ruff 2 and 7 Test, Animal Naming, HVLT-R, Medical College of Georgia Complex Figure Test and COWAT and Animal Naming test.	Non-significant
Komulainen et al., 2021 [31]	Single-blind, parallel	Finland	Apparently cognitively healthy	469	57–78	Finnish Nutrition Recommendations	General recommendations	4 y	CERAD neuropsychological battery, MMSE	Non-significant
Tsolaki et al., 2020 [29]	Double-blind, parallel	Greece	MCI	54	69.8 ± 6.9	G1: 50 mL/d extra virgin olive oil + Mediterranean diet G2: 50 mL/d high phenolic early harvest extra virgin olive oil + Mediterranean diet	Mediterranean diet without olive oil	12 m	ADAS-Cog, DST, Letter fluency, MMSE, Rivermead Behavioral Memory Test-Story Recall, ROCF, TMT-A and TMT-B, CDT	ADAS-cog, Digit Span and Letter fluency. MMSE
Wade et al., 2019 [27]	Blinding information not available, crossover	Australia	Cognitively healthy	35	61.0 ± 7.1	Mediterranean diet with 2–3 weekly servings of fresh, lean pork	Low fat diet	8 w	Primary: Blood pressureSecondary: Cantab, ACE-R	Processing speed
Wade et al., 2020 [28]	Single-blind, crossover	Australia	Cognitively healthy	43	60.2 ± 6.9	Mediterranean diet: 3–4 daily servings of dairy foods	Low fat diet	8 w	Primary: Blood pressure Secondary: Cantab, ACE-R	Processing speed.
Food-based interventions										
Edwards et al., 2019 [33]	Single-blind, parallel	USA	Cognitively healthy	163	25–45	Fresh Hass Avocado: male (175 g/d containing 701 µg lutein and zeaxanthin); female (140 g/d containing 561 µg lutein and zeaxanthin)	Isocaloric meal without avocado (lutein/zeaxanthin content was 164/205 µg/d)	12 w	Flanker, Nogo tasks, Oddball Task, Kaufman Brief Intelligence Test	Accuracy in the Flanker task (attentional inhibition)
Kuroda et al., 2019 [34]	Blinding information not available, parallel	Japan	Cognitively healthy	52	72.9 ± 0.8	100 g/d of ultra-high hydrostatic pressurizing brown rice	100 g/d of white rice	24 m	HDS-R, MMSE, FAB, Cognitive CADi	HDS-R
Sakurai et al., 2020 [32]	Double-blind, parallel	Japan	Cognitively healthy	61	60–84	3.0 g/d Matcha new green tea powder	Black tea flavored powder	12 w	MoCa, MMSE, WMS-DR	MoCA score in the active group of women
Sala-Vila et al., 2020 [37]	Single-blind, parallel	Spain, USA	Cognitively healthy	708	63–79	Diet with walnuts (15% of energy)	Regular diet free of nuts	2 y	RAVLT, ROCF, SVF, BNT, VOSP, WAIS-III, TMT A and B, Phonemic Fluency, SCWT, SDMT, DST, CPT-II, fMRI	Perception score
Sandberg et al., 2018 [36]	Blinding information not available, crossover	Sweden	Apparently cognitively healthy	43	63.6 ± 5.3	Ased bread consisted of a whole grain rye kernel/flour mixture (1:1 ratio) supplemented with resistant starch type 2 (239.2 g/d)	White wheat flour bread (170.9 g/d)	3 d	VWM test, selective attention test.	Non-significant
Suzuki et al., 2019 [38]	Blinding information not available, crossover	Japan	MCI	71 (F)	≥70	33.4 g/d Camembert cheese	33.4 g/d of processed cheese made from mozzarella cheese and cream cheese	3 m	Primary: Serum brain-derived neurotrophic factor concentration Secondary: MMSE	Non-significant
Uenobe et al., 2019 [35]	Crossover, blinding information not available	Japan	Apparently cognitively healthy	31	IG 84.3 ± 0.3 CG 83.8 ± 9.1	Dewaxed brown rice (3 meals/d, daily dose not available)	white rice (3 meals/d, daily dose not available)	6 m	HDS-R	HDS-R
Dietary supplement interventions										
Abe et al., 2020 [81]	Single-blind, parallel	Japan	Apparently cognitively healthy	64	85.5 ± 6.8	G1: 1.2 g/d L-leucine and 20 μg/d cholecalciferol and 6 g/d medium-chain triglycerides G2: 6 g/d medium-chain triglycerides	6 g/d of long-chain triglycerides	3 m	Primary: 10-s leg open and close test (muscle function) Secondary: MMSE, NM scale	MMSE
Ahles et al., 2020 [51]	Double-blind, parallel	Netherlands	Apparently cognitively healthy	101	40–60	Aronia melanocarpa extract: 90 and 150 mg/d (16 mg and 27 mg anthocyanins respectively)	Placebo (maltodextrin 150 mg/d)	24 w	SCWT, grooved pegboard test, number cross-out test	Psychomotor speed.
Arellanes et al., 2020 [71]	Double-blind, parallel	USA	Cognitively healthy	33	58–90	2152 mg/d of DHA and vitamin B complex(1 mg vitamin B12, 100 mg of vitamin B6 and 800 µg of folic acid)	Placebo(corn/soy oil and vitamin B complex (1 mg/d vitamin B12, 100 mg/d of vitamin B6 and 800 µg/d of folic acid))	6 m	Primary: DHA cerebral fluid contendingSecondary: CVLT-II, MoCa, TMT-A, TMT-B, CDR, MRI	Non-significant
Baleztena et al., 2018 [86]	Double-blind, parallel	Spain	Apparently cognitively healthy	99	86.9 ± 5.9	DHA 750 mg, EPA 120 mg, vitamin E 15 mg, Ginkgo biloba 180 mg, phosphatidylserine 45 mg, tryptophan 285 mg, vitamin B12 E 15 mg, folate 750 mg, daily	Placebo (gelatin capsule)	1 y	MMSE, GDS, SPMSQ, SVF, CDT	Non-significant
Ban et al., 2018 [84]	Double-blind, parallel	South Korea	SCD	75	40–65	*Tremella fuciformis* capsules (Low-dose 600 mg/d, high-dose 1200 mg/d)	Placebo	4, 8 w	SMCQ, WCST, Self-reported cognitive impairment, CANTAB	Short-term memory, executive functions, Gray matter volumes of brain regions Subjective memory
Bensalem et al., 2019 [53]	Double-blind, parallel	France and Canada	Cognitively healthy	215	64.66 ± 2.91	600 mg/d of polyphenol-rich extract from grape and blueberry (258 mg flavonoids)	600 mg/d of pure maltodextrin	6 m	CANTAB (PALTEA section), CANTAB (SSP, VRM sections)	Verbal episodic memory in subjects with the highest cognitive impairments
Boyle et al., 2019 [80]	Double-blind, parallel	UK	Apparently cognitively healthy	54 (M)	18–55	250 mL/d water-based drink produced with milk protein concentrates (2.7 g of phospholipids and 300 mg phosphatidylserine)	Placebo (drink matched by adding butteroil)	6 w	2-back and attention switch task	Attention switch task
Carmichael et al., 2018 [65]	Double-blind, crossover	USA	Cognitively healthy	11	67.3 ± 2.01	Liquid emulsification (16 mg/d total ginsenosides: 960 mg EPA, 624 mg DHA, and 26 mg of green tea catechins)	Corn oil using non-essential fatty acid without ginsenosides nor catechins	26 d	MMSE, DSST, Stroop test and LM (I & II), MRI	MMSE, Stroop test, DSST; brain activation (anterior and posterior cingulate cortex), functional connectivity (middle frontal gyrus and anterior cingulate cortex)
Chai et al., 2019 [54]	Blinding information not available, parallel	USA	Cognitively healthy	37	65–80	480 mL tart cherry juice	Unsweetened black cherry flavored	12 w	CANTAB (PAL, RVP, RTI, SWM and DST sections)	Memory, learning tasks, sustained attention, spatial working memory
Chhetri et al., 2018 [62]	Double-blind, parallel	France	Cognitively healthy	637	75.3 ± 4.4	PUFA ω-3 supplementation (800 mg/d of DHA and ≤225 mg/d of EPA)	Placebo (paraffin oil)	3 y	Composite score (FCSRT, MMSE, DSST, CNT), TMT, WAIS-R and COWAT.	Non-significant
Chupel et al., 2018 [42]	Parallel, blinding information not available	Portugal	Apparently cognitively healthy	24 (F)	83.5 ± 6.9	150 mL/d water with 1.5 g taurine	Usual care	14 w	MMSE	Non-significant
Cox et al., 2020 [52]	Double-blind, parallel	Australia	Cognitively healthy	89	50–85	80 mg/d of curcumin extract (Longvida©)	Placebo	12 w	DATT, vMWM, Serials Subtractions, Arrow Flankers Task	Working memory
Danthiir et al., 2018 [66]	Double-blind, parallel	Australia	Cognitively healthy	403	65–90	1720 mg/d DHA and 600 mg/d EPA (plus mixed tocopherols added as antioxidant, 2.8–4.5 mg/g)	Placebo (3960 mg/d of low polyphenol olive oil + 40 mg/d of fish oil (7.2 mg EPA and 4.8 mg DHA))	18 m	Fluency, working memory, reasoning, and short-term memory (word memory—immediate recall), speed of memory scanning (Sternberg’s number and letter memory scanning), odd-man-out reaction time, perceptual speed, inhibition, simple and choice reaction time, psychomotor speed, MMSE score	Perceived cognitive mistakes Psychomotor speed only in men
Decroix et al., 2019 [50]	Double-blind, crossover	Belgium	Cognitively healthy	22	41.2 ± 15.8	900 mg Cocoa Flavanols dissolved in 300 mL of skimmed milk.	15 mg Cocoa Flavanols dissolved in 300 mL of skimmed milk.	1 d	Flanker test and fMRI	Reaction time (flanker test) The BOLD response
Delrieu et al., 2020 [64]	Double-blind, parallel	France	Apparently cognitively healthy	67	76.4 ± 4.2	800 mg/d of V0137 (DHA + EPA)	Placebo	12 m	(^18^F) FDG PET imaging, FCRST, COWAT, Category Naming Test, DSST of WAIS-R, TMT, MMSE, CDR and Z-score, MRI	Non-significant
Fukuda et al., 2020 [82]	Double-blind, parallel	Japan	SCD	60	45–64	35 mg/d of matured hop bitter acids (MHBAs)	Placebo (dextrin capsules)	12 w	Word recall test; story recall test, VFT; semantic and phonemic fluency task; WMS-R; SWM; Stroop test, subjective memory performance, TMT-A, TMT-B	Verbal fluency and Stroop test
Fukuda, et al., 2020 [83]	Double-blind, parallel	Japan	SCD	100	45–69	35 mg/d of MHBAs	Placebo (dextrin)	12 w	CAT, SDMT, memory updating test; Position response test; Memory (RAVLT, S-PA, WMS-R)	Memory retrieval and attention
Giudici et al., 2020 [61]	Double-blind, parallel	France	Cognitively healthy	715	75.3 ± 4.4	PUFA ω-3 (800 mg/d of DHA and ≤ 225 mg/d of EPA)	Placebo (paraffin oil)	3 y	Z-score (MMSE, Digit Symbol Substitution Test, free and total recall of the Free and Cued Selective Reminding test, and Category Naming Test)	Non-significant
Hamasaki et al., 2019 [41]	Double-blind, parallel	Japan	Apparently cognitively healthy	35	≥50	3 tablets/d of lactotripeptide (casein hydrolysate with 1.4 mg valine-proline-proline and 2.0 mg isoleucine-proline-proline)	Placebo (sodium caseinate)	8 w	Primary: Oxygenated hemoglobin (oxy-Hb) concentration (oxy-Hb change) in the prefrontal cortex during the Stroop task Secondary: Stroop task and stroop interference time	Non-significant
Herrlinger et al., 2018 [55]	Double-blind, parallel	USA	SCD	90	59.4 ± 0.6	600 mg/d or 900 mg/d spearmint extract	Placebo	90 d	Cognitive Drug Research System	Quality of working memory, spatial working memory accuracy, vigor-activity, alertness and behavior following wakefulness after the highest dose
Hu et al., 2018 [47]	Double-blind, parallel	China	MCI	181	65–75	400 IU/d vitamin D	Placebo (starch granules)	1 y	Chinese version of the WAIS-R; MMSE	Scores of information, digit span, vocabulary, block design and picture arrangement
Hwang et al., 2019 [56]	Double-blind, parallel	South Korea	MCI	100	55–85	800 mg/d of Lactobacillus plantarum C29-fermented soybean enriched with isoflavones and saponins.	Placebo (cellulose)	12 w	Computerized neurocognitive function tests, Verbal learning Test, ACPT, DST	cognitive functions and attention
Igase et al., 2017 [74]	Double-blind, parallel	Japan	Apparently cognitively healthy	84	71 ± 9	125 mL juice enriched with 6 mg/d auraptene	125 mL juice enriched with 0.1 mg auraptene/day	24 w	MCI Screen using 10-word immediate recall test, MMSE	10-word immediate recall test
Ito et al., 2018 [73]	Double-blind, parallel	Japan	MCI	21	57–78	6 mg/d astaxanthin and 10 mg/d sesamin	Placebo (filling agent such as safflower oil, starch and water, dispersants and artificial colorants)	12 w	CNSVS (SDC, Stroop test, shifting attention test and continuous performance test domains), Japanese version of the ADAS-cog	Psychomotor speed and processing speed
Kita et al., 2018 [44]	Double-blind, parallel	Japan	SCD	101	45–64	1 g/d whey peptide which included 1.6 mg of glycine–threonine–tryptophan–tyrosine peptide	Placebo (maltodextrin)	12 w	Story recall tests, VFT, Hamamatsu Higher Brain Function Scale, Japanese version of the RBMT, Stroop test, DST, and paced auditory serial addition test	VFT, Stroop, subjective memory function in subjects with high-level fatigue
Kuszewski et al., 2020 [68]	Double-blind, parallel	Australia	Cognitively healthy	152	50–80	G1: Fish oil (2000 mg/d DHA + 400 mg/d EPA); G2: curcumin (160 mg/d); G3: Fish oil (2000 mg/d DHA + 400 mg/d EPA) and curcumin (160 mg/d)	Placebo (mix of corn and olive oil with 20 mg fish oil)	16 w	Primary: Cerebrovascular function Secondary: 7 tests from the NIH toolbox, RAVLT, Forward Spatial Span Test, and TMT parts A and B.	Processing speed (males in G1) verbal memory (males in G2)
Lau et al., 2020 [59]	Double-blind, parallel	Malaysia	MCI	36	66.42 ± 0.63	500 mg/d Biokesum (extract of Persicaria minor which contains quercetin-3-glucuronide (not less than 0.45%), quercitrin (not less than 0.15%)) and total phenolic content (not less than 100 mg GAE/g dE).	Placebo (560 mg/d maltodextrin)	6 m	MMSE, DST, RAVLT, VR I-II, Digit symbol substitution, POMS, fMRI, N-Back.	VR II, bilateral dorsolateral prefrontal cortex activation
Li et al., 2018 [70]	Double-blind, parallel	China	MCI	240	≥60	FA + DHA (FA 800 μg/d + DHA 800 mg/d) FA (FA 800 μg/d); DHA (DHA 800 mg/d)	Placebo tablets (corn starch) and placebo capsules (soybean oil)	6 m	FSIQ and Chinese version of WAIS-R which included three verbal subtests (information, arithmetic and digit span) and three performance subtests (block design, picture completion and picture arrangement)	FSIQ, arithmetic, digit span, picture completion and block design (FA +DHA group) digit span and block design (FA group) information, arithmetic and digit span (DHA group) FSIQ scores, arithmetic and picture completion (FA+ DHA vs. FA) picture completion and block design (FA+ DHA vs. DHA)
Ma et al., 2019 [49]	Single-blind, parallel	China	MCI	240	≥ 65	Folic acid alone (two tablets of 400 μg/d of folic acid) Vitamin B12 alone (one tablet with 25 μg/d vitamin B12) Folic acid plus vitamin B12 (two tablets of 400 μg/d folic acid plus one tablet with 25 μg/d vitamin B12)	Without treatment	6 m	Chinese version of the WAIS-RC (Information, Similarities, Vocabulary, Comprehension, Arithmetic, DST, Block Design, Picture Completion, Digit Symbol-Coding, Object Assembly, and Picture Arrangement domains), IQ index, MMSE.	Full Scale IQ, verbal IQ, Information and Digit Span
Marriott et al., 2021 [60]	Double-blind, parallel	USA	Apparently cognitively healthy	555	20–35	8 dietary supplements daily of krill oil (2.3 g ω-3)	Placebo capsules (macadamia nut oil)	20 w	Stroop Color-Word Inhibition Test, SDMT, Figural Continuous Paired Associates Test	Non-significant
Masuoka et al., 2019 [39]	Double-blind, parallel	Japan	MCI	54	49–86	750 mg/d anserine plus 250 mg/d carnosine	Placebo	12 w	CDR, ADAS-cog test, WMS-1, WMS-2, MMSE, gloCDR	gloCDR score in APOE4(+)
McNamara et al., 2018 [69]	Double-blind, parallel	USA	SCD	94	62–80	Fish oil group (FO): 1.6 g/d EPA and 0.8 g/d DHA Blueberry group (BB): 25 g/d blueberry powder rich in anthocyanins FO + BB group: 1.6 g/d EPA, 0.8 g/d DHA, 25 g/d blueberry powder	Placebo (corn oil powder)	24 w	TMT A and B, Controlled Oral Word Production, Alternate forms of HVLT-R, Dysexecutive Questionnaire.	Memory discrimination (BB)
										Cognitive symptoms (BB and FO)
Ochiai et al., 2019 [58]	Double-blind, crossover	Japan	MCI	34	73.7 ± 6.0	1.107 g/d of chlorogenic acid (CGA) (553.6 mg in 100 mL of water)	Placebo	12 w	MMSE; ADAS-cog test; TMT-A, TMT-B); cognitive function tests.	Number of errors in the TMT-B test
Owusu et al., 2019 [46]	Double-blind, parallel	USA	SCD	260 (F)	68.2	D3 vitamin daily dose: 2400 IU, 3600 IU, or 4800 IU.	Placebo (1200 mg/d calcium carbonate)	3 y	MMSE	Non-significant
Perry et al., 2018 [78]	Double-blind, parallel	United Kingdom	Cognitively healthy	44	61 ± 9.26	10 mL/d SRM ethanol extract (SRM: Salvia officinalis L., Rosmarinus officinalis L. and Melissa officinalis L., collected and individually extracted 0.5 g/mL in 45% EtOH)	Placebo (50%º fresh sweet cicely)	2 w	Immediate and delayed word recall to assess verbal working and episodic memory, VWM, Verbal Episodic Memory.	Delayed word recalls (subjects under 63 years)
Reid et al., 2018 [77]	Double-blind, parallel	South Korea	Cognitively healthy	60	IG 72.35 ± 5.54 CG 74.57 ± 5.69	1.5 g/day of fermented Laminaria japonica	Placebo (1.5 g/d of sucrose pill)	6 w	MMSE, Numerical Memory Test, Raven’s Standard Progressive Matrices, Flanker Test, Iconic Memory Test and TMT.	K-MMSE, flanker test scores, working memory, visual and spatial reasoning
Saitsu et al., 2019 [76]	Double-blind, parallel	Japan	Cognitively healthy	34	>50	3.2 g/d Fruiting body of *Hericium erinaceus*	Placebo	12 w	MMSE, Benton visual retention test, and S-PA.	MMSE
Schönhoff et al., 2018 [43]	Double-blind, crossover	Germany	Cognitively healthy	20	35 ± 14	125 mg/d L-homoarginine supplement	Placebo	4 w	VLMT, TAP, TMT, FWIT, Regensburg verbal fluency test	Non-significant
Schwarz et al., 2018 [72]	Double-blind, parallel	Germany	MCI	49	50–80	1320 mg EPA, 880 mg DHA, 15 mg vitamin E, daily	Sunflower oil	26 w	MRI	Cerebral perfusion
Stavrinou et al., 2020 [67]	Double-blind, parallel	Cyprus	MCI	46	≥65	20 mL/d (4.14 g DHA, 0.810 mg EPA, 1.8 g GLA, 3.15 g LA, 0.6 mg vitamin A, 22 mg α-tocopherol, 760 mg ɣ-tocopherol)	Placebo (20 mL pure virgin olive oil)	6 m	ACE-R, MMSE, TMT, Stroop Color and Word Test, symbol cancellation test.	Functional capacity, physical Health
										Time to complete ACE-R, MMSE and STROOP
Tabue-Teguo et al., 2018 [63]	Double-blind, parallel	France	Cognitively healthy	843	75.3 ± 4.4	800 mg/d of V0137 (DHA + EPA)	Placebo (paraffin oil)	3 y	COWAT, Category Naming Test, DSST of the WAIS-R, TMT, MMSE	Non-significant
Tadokoro et al., 2019 [87]	Double-blind, parallel	Japan	MCI	78	65–85	Tablet daily dose: coenzyme Q10 (10 mg), niacin amid (2 mg), L-cystine (50 mg), ascorbic acid (94 mg), succinic acid (10 mg), fumaric acid (10 mg), L-glutamine (85 mg), and riboflavin (4 mg).	Placebo	6 m	MMSE and HDS-R	MMSE, HDS-R
Tohda et al., 2020 [75]	Double-blind, parallel	Japan	Cognitively healthy	31	40–80	203.01 mg/d of *Eleutherococcus senticosus* leaf extract, 20.01 mg/d of Rhizome of *Drynaria fortunei* extract, 133.38 mg/d crystal cellulose and 3.60 mg/d calcium stearate	Placebo (356.4 mg crystal cellulose and 3.60 mg calcium stearate)	12 w	RBANS, Japanese versions of MMSE	Figure recall sub score of RBANS (language domain, semantic fluency and figure recall)
Varanoske et al., 2018 [45]	Double-blind, parallel	USA	Apparently cognitively healthy	19 (M)	18–35	12 g/day β-Alanine	Placebo	14 d	Visual analog scale (Mood), Serial Subtraction Test (Mathematical processing), reaction time (by using Dynavision D2 Visuomotor Training Device), visual tracking speed (by using Neurotracker multiple object tracking device)	Errors on reaction time testing
Wirth et al., 2018 [85]	Double-blind, parallel	Germany	SCD	30	65.6 ± 6.2	750 mg/d of spermidine-rich plant (1.2 mg of the natural polyamine plus 510 mg cellulose)	Placebo (750 mg/d of potato starch and 510 mg/d cellulose)	3 m	Memory performance (Behavioral non-verbal MST), AVLT, digit symbol substitution test	Non-significant
You et al., 2021 [57]	Double-blind, parallel	Malaysia	MCI	48	65.11 ± 4.05	500 mg/d of *Cosmos caudatus* powder	Placebo (1000 mg/d of maltodextrin)	12 w	MMSE, DS, RAVLT, Digit Symbol substitution, Visual Reproduction and POMS	MMSE
Yuda et al., 2020 [40]	Double-blind, parallel	Japan	Cognitively healthy	268	≥40	200 μg/d Met-Lys-Pro peptide in 1 g casein hydrolysate	Placebo (1 g dextrin with no detectable MKP)	24 w	ADAS-cog, HDS-R, MoCa Japanese version, and Short-Form Health Survey (SF-8).	Orientation in ADAS-cog
Zajac et al., 2020 [48]	Double-blind, parallel	Australia	Cognitively healthy	436	60–90	Vitamin D2-enrich Mushroom (600 µg/d D2 vitamin); Vitamin D3 (600 µg/d D3 vitamin), standard mushroom capsule (NA)	Placebo	6 m	CSIRO Cognitive Assessment Battery (C-CAB)	Non-significant
Zhu et al., 2020 [79]	Double-blind, parallel	USA	Apparently cognitively healthy	250	40–85	Magnesium glycinate capsules	Placebo (microcrystalline cellulose)	12 w	MoCa score	MoCa

^1^ ACE-R, Addenbrooke’s Cognitive Examination-Revised; ACPT, auditory continuous performance test; ADAS-cog, Alzheimer’s Disease Assessment Scale-cognitive component; AVLT, Auditory Verbal Learning Test; Behavioral non-verbal MST, mnemonic similarity task; BNT, Boston Naming Test; CADi, Cognitive Assessment for Dementia iPad version; CANTAB, Cambridge Neuropsychological Test Automated Battery; CAT, Clinical Assessment for Attention; CDR, Clinical Dementia Rating; CDT, Clock Drawing Test; CERAD, Consortium to Establish a Registry for Alzheimer’s Disease; CNSVS, Central Nervous System Vital Signs; CNT, computerized neurocognitive test; COWAT, Controlled Oral Word Association Test; CPT-II, Conner’s Continuous Performance Test-II; CSIRO C-CAB, CSIRO Cognitive Assessment Battery; CVLT-II, California Verbal Learning Test 2nd edition; DASH, Dietary Approaches to Stop Hypertension; DATT, Divided Attention Tracking Task; DSST, Digit Symbol Substitution Test; DST, Digit Span test; FAB, Frontal Assessment Battery; FCSRT, Free and cued selective reminding test; fMRI, Functional magnetic resonance imaging; FSIQ, Full Scale IQ subset; FWIT, Farb Wechsler Interferenz Test; GDS, Global Deterioration Scale; gloCDR, Global clinical dementia rating; HDS-R, Hasegawa’s Dementia Scale-revised; HVLT-R, Hopkins Verbal Learning Test-Revised; LM (I and II), Logical Memory I and II; MCI, Mild Cognitive Impairment; MHBA, Matured Hop Bitter Acids; MMSE, Mini-mental state examination; MoCA, Montreal Cognitive Assessment; NM, Nishimura geriatric rating scale for mental status; PAL, Paired Associate Learning; PALTEA, Cantab Paired Associate Learning total errors adjusted; POMS, Profile of Mood State; RAVLT, Rey Auditory Verbal Learning Test; RBANS, Repeatable Battery for the Assessment of Neuropsychological Status; RBMT, Rivermead Behavioral Memory Test; ROCF, Rey-Osterrieth Complex Figure; RTI, Reaction time; RVP, Rapid visual information processing; SCD, Subjective Cognitive Decline; SCWT, Stroop Color Word Test; SDC, symbol digit coding test; SDMT, symbol digit modalities test; SMCQ, Subjective Memory Complaints Questionnaire; S-PA, Standard verbal paired associate learning test; SPMSQ, Short Portable Mental Status Questionnaire; SSP, Spatial Span; SVF, Semantic Verbal Fluency Test; SVF, Semantic Verbal Fluency; SWM, Spatial Working Memory; TAP, Test of Attenuance Performance; TMT, Trail Making Test; VFT, Verbal Fluency Test; VLMT, verbal learning memory test; vMWM, Virtual Morris Water Maze; VOSP, Visual Object and Space Perception battery; VR, Visual reproduction test; VRM, Verbal episodic and recognition memory; VWM, Verbal working memory; WAIS-R, Wechsler Adult Intelligence Scale-Revised; WCST, Wisconsin Card Sorting Test; WMS, Wechsler memory scale.

### 3.6. Risk of Bias Assessment

In terms of quality assessment, 48% of the studies were judged at low risk of bias, 13% with some concerns and 39% at high risk of bias (see Figure 2, Figure 3 and Figure 4). Many of the low-quality studies did not inform on the randomization process or on the blinding of outcome assessors, especially regarding diet counselling and food-based interventions. Some others did not evaluate adherence to the intervention or presented deviations from the intended interventions. Notably, only 39% of the studies followed an intention-to-treat assessment (Appendix A), whereas 61% followed a per-protocol assessment (Appendix A), which is generally considered a high-risk factor for bias. Moreover, studies with a small sample size were usually assessed at high risk of bias, whereas large sample sizes were often judged to be at low risk of bias (Figure 5).

## 4. Discussion

The aim of the present systematic review of RCTs was to update and provide the most comprehensive analysis to date of the effectiveness of dietary patterns, specific foods and nutritional supplements on cognitive health outcomes in subjects without symptoms of dementia. Overall, significant improvement in cognitive functions was observed in most (n = 44) but not all (n = 17) studies, irrespective of treatment type (diet, food, supplement), duration or cognitive status (cognitively healthy subjects or cognitively impaired older adults).

Diet has been suggested to have an important role in cognitive health and the development of dementia [88]. However, most of the intervention studies conducted to date mainly assessed the effect of dietary supplements on cognitive function, while few trials were designed to include specific foods or dietary patterns. Within the dietary patterns, the Mediterranean diet is probably the most investigated and with the most promising results [89]. A sub-analysis previously conducted in the framework of the PREDIMED trial showed that following a Mediterranean diet enriched with olive oil or nuts improved global cognition, memory and attention after 4 years of follow-up [90]. Since 2018, 3 RCTs evaluating the effect of following a Mediterranean diet enriched with either fresh lean pork (MedPork study) or dairy products (MedDairy study) or high phenolic early harvest extra virgin olive oil (HP-EH-EVOO) in cognitively healthy subjects have been conducted, showing increased processing speed as compared to a low-fat diet [27,28] and better scores on global cognition, attention and fluency tasks compared to a Mediterranean diet not supplemented with high-phenolic EVOO [29]. However, because in the MedPork [27] and MedDairy [28] studies the control groups followed a low-fat diet instead of a Mediterranean diet, it cannot be elucidated whether the observed benefits were due to the synergistic role of the healthy components of the Mediterranean diet, rather than the addition of fresh pork or dairy products. Similarly, in the DR’s EXTRA study, no difference in the 4-year change in global cognition was observed between the diet group and the control group, and this could be due to the fact that diet quality also improved in the control group, which could confound the effect of a healthy diet [31]. In addition, the DASH diet, with similar dietary composition to the Mediterranean diet, but with a higher intake of low-fat dairy products, potassium and less sodium, was shown in previous observational studies to be associated with improved cognition and lower risk of cognitive impairment [91]. This is probably due to the fact that the DASH diet shares similar characteristics with the Mediterranean diet, in addition to reduced sodium intake, which is associated with improved blood–brain barrier integrity [92,93]. The lack of significant effects in Blumenthal et al., might be due to the low number of participants in the intervention group and/or the relatively modest adherence to the diet regime [30].

Some key foods of the Mediterranean diet, such as EVOO, but also other plant-based foods including brown rice, matcha green tea and avocado, have also shown significant improvements in different cognitive domains such as speed of execution [34], orientation and language [34,35], and attention and global cognition [33,34,35]. Surprisingly, the Walnuts and Healthy Aging study, conducted at two centers, found no consistent effect of walnut consumption on global cognition in all participants. However, when data was analyzed separately by recruitment centers, an improvement in global cognition and perception scores was observed in participants recruited in Barcelona but not in the Loma Linda subjects. These differences between centers were potentially attributed to lower levels of education and lower plasma α-linolenic acid concentrations in the Barcelona participants at baseline [37]. Animal-based foods, such as camembert cheese, worsened cognition, even when levels of the neuroprotective BDNF were high [38], but, in contrast, intake of bovine milk-derived phospholipid was shown to protect cognitive performance under stress in terms of better sustained attention. Taken together, the effects of phospholipid supplementation may be limited to subjects with high levels of stress, as they attenuate stress response mediated by the HPA axis [80].

It is important to note that all but one study [37] included a relatively small number of participants, between 20 to 45 subjects, with a medium follow-up (<12 weeks), showing a high risk of bias. The studies on dietary counselling and food-based interventions included in this systematic review were conducted in cognitively healthy subjects and in patients with MCI, showing similar improvements in both groups, which cannot be extrapolated to adults with subjective cognitive decline.

Most of the RCTs included in this systematic review (48 out of 61) have evaluated the effect of dietary supplements (mostly plant-based supplements) on cognitive health using parallel study designs and different intervention durations (maximum 3 years). Half of the interventions evaluating dietary supplements (24 out of 48) were carried out for more than 12 weeks, 9 studies lasted less than this period and 16 studies were conducted for exactly 12 weeks.

### 4.1. Protein and Amino Acid Supplements

Met-Lys-Pro peptide, whey peptide and β-alanine significantly improved cognitive function [40,44,45]. Interestingly, the effects of whey peptide at week 12 were reduced compared to week 6 [44]. Some studies have suggested that this reduction is due to the learning effects on the tests, especially the less complex ones, resulting in smaller differences between the placebo and intervention groups. Still, the consumption of L-homoarginine and lactotripeptide did not exert any beneficial effect, which could be attributed to the short supplementation period of 4 and 8 weeks, respectively, and/or the small sample size (n = 20 and n = 35) [41,43]. Taurine is a sulfur compound with many functions in the nervous system. Recent data from animal studies suggest that taurine can improve cognitive function [94]. Considering that its plasma concentration decreases in older people, the lack of effect of taurine might be due to the small sample size (n = 24) or that a sufficient plasma taurine concentration was not reached to exert any beneficial effect [42]. Finally, anserine and carnosine only improved cognition in APOE4+ subjects, but not in the general population. Whether this over-stratification provokes false positive or there is an interaction between APOE4 and nutritional interventions deserves further attention [39].

### 4.2. Vitamin Supplements

The association of low vitamin D with global cognitive deficits is evident in several cross-sectional and longitudinal studies, but causality remains unclear [95]. The 3 RCTs included in this systematic review did not contribute to clarifying the role of vitamin D in cognitive decline. Thus, while vitamin D supplementation did not show beneficial effects in studies in cognitively healthy participants [46,48], it did in patients with MCI [47]. It is possible that the effects of vitamin D only become apparent when cognitive impairment is present. As for vitamin B12, it may exert a neuroprotective effect on cognitive function [49], alone or in combination with folic acid. The lack of effect was observed in one study included in our systematic review and this may be due to the use of diabetic medication in the study participants, including ACE inhibitor, angiotensin receptor blocker or metformin, which were also found to be neuroprotective [96]. Whether this over-stratification provokes false positive or there is an interaction between APOE4 and nutritional interventions deserves further attention [39].

### 4.3. Polyphenol Supplements

In general, polyphenol supplementation using extracts of fruits rich in anthocyanins such as blueberries, Montmorency tart cherries and grapes, as well as cocoa flavanols, Aronia melanocarpa, curcumin, spearmint, Lactobacillus plantarum C29-fermented soybean, Cosmos caudatus, chlorogenic acid and the nutraceutical Biokesum, have shown significant reduction in age-related decline in memory and cognition that could be explained by their antioxidant and anti-inflammatory properties [50,51,52,53,54,55,56,57,58,59]. However, a crossover trial published in 2016 used a single 60 mL dose of Montmorency tart cherry concentrate for 14 days and failed to show any beneficial effect on cognition, including attention, working memory, inhibition and cognitive flexibility in middle-aged participants, despite improvements in certain vascular functions [97]. These contrasting results could be partially explained by the lower doses used in this study compared to another study included in this review in which elderly participants consumed 2 cups of Montmorency tart cherry juice over a longer period (12 weeks) [54].

### 4.4. PUFAs Supplements

Despite the well-known role of omega-3 fatty acids in improving vascular endothelial function [98], the effect of omega-3 fatty acid supplementation on cognitive function has provided mixed results [99,100] and showed no improvement in the 5 RCT included in this review. Even if the lack of beneficial effect in the trial by Giudici et al. [61] might be due to decreasing adherence over time, similar results were obtained in the trial by Chhetri et al. [62] leading to the suggestion that a multidomain intervention might be more useful than PUFAs alone. However, in The Multidomain Alzheimer Preventive Trial (MAPT), no effects on brain glucose metabolism [64] or cognitive function were observed in subjects with SCD [63]. No beneficial effect was also observed in young men [60]. This is consistent with a previous RCT, in which an increase in oxyhemoglobin concentration was observed during a mental arithmetic task in participants assignedfiveto krill oil supplementation, but no changes in behavioral performance were reported [101].

However, contrary to these results, Cook et al. [102] reported that cognitively healthy young women with lower omega-3s had lower attention scores. Therefore, discrepancies between these studies might be due to gender differences in absorption and/or response to omega-3 fatty acid supplementation. Additionally, the poor retention rate of only 44.1% and the questionable compliance of the participants who completed the study make the results underpowered to test the alternative hypotheses.

### 4.5. Combination of PUFAs, Polyphenols and Vitamins

The synergistic action of antioxidants and anti-inflammatory bioactive compounds may also exert beneficial effects on some cognitive functions, such as executive [65], attention [65], working memory [65], intelligence [70], verbal memory and processing speed [68]. However, this synergistic action did not exert a beneficial effect in all studies. In this context, a combination of fish oil and blueberry extracts did not improve cognition [69]. One explanation is that long-term, daily supplementation with the combination treatments may lead to an over-regulation of the transcription factor Nrf2, whose excessive activation could contribute to adverse effects [103]. Additionally, for n-3 FA supplementation to exert an effect on cognitive outcomes, it is likely that adequate brain levels of n-3 FA must be reached. In the study by Arellanes et al. [71], vitamin B complex and n-3 FA were combined, but their levels could not be properly measured, so the lack of effects observed in this study may be due to inadequate brain delivery. Nevertheless, omega-3 FA may improve perfusion in cerebral regions typically affected in AD, which would help maintain brain structure and function and potentially delay conversion to dementia [72]. It is also possible that fish oil supports cognitive function in the short term, but that the positive effects may not apply over longer periods [66].

### 4.6. Other Types of Dietary Supplements

Other dietary supplements that have shown significant improvements on different cognitive domains are fungi and algae extracts (Hericiumerinaceus, Tremella fuciformis and Laminaria japonica), other plant extracts (Eleutherococcussenticosus leaf, Drynariafortuneirhizome, combined sage, rosemary and melissa, auraptene, astaxanthin and matured hop bitter acids), minerals such as magnesium, medium-chain triglycerides and the complex Twendee X (supplement containing a mix of eight antioxidants). The consumption of the above antioxidant and anti-inflammatory supplements not only significantly improved subjective memory complaints [84] and scores in different cognitive domains such as psychomotor and processing speed [73], working memory [76,77,79,81], short-term memory [74,76,77,84], spatial working memory [75,79,81,87], verbal episodic memory [78], verbal fluency [75,82,83], language [75,79,81,87], executive functions [79,82,84] and attention [79,83,87], but also cortical gray matter integrity in AD-related brain regions [84]. The spermidine-rich plant extract also had an impact on memory performance in subjects with SCD, although this was not significant, probably due to the small sample size (n = 30) and/or the short follow-up period (3 months) [85]. Differences in cognitive outcomes were also dependent on the patients’ nutritional status. Well-nourished subjects showed a significant improvement in memory following multi-nutrient supplementation, the main component of which was omega-3 PUFA, but this improvement was not apparent in those with poorer nutritional status [86]. This might be explained by a combination of the potential synergies of the nutrients in the supplement and the good nutritional status [104]. Similarly, whereas antioxidant vitamins C and E were reported to have little effect on cognition [105,106], the combination of antioxidants in Twendee X showed stronger antioxidant and anti-inflammatory effects than single antioxidant vitamins in MCI subjects [87].

### 4.7. Potential Mechanisms

Oxidative stress and inflammation are common features in age-related cognitive decline [107]. In this vein, recent studies performed in cognitively normal elderly subjects have shown that levels of total antioxidant capacity are associated with decreased cortical integrity, poorer cognitive function, and increased cerebral amyloid beta burden [108,109], the earliest neuropathological event in AD. Therefore, antioxidants and anti-inflammatory agents are of particular interest in the modulation of reactive oxygen species (ROS) and inflammatory biomarkers [110] (Figure 6).

Plant-based food interventions, particularly those using colorful fruit and vegetable-bearing plant foods, are rich in polyphenols, anthocyanins, carotenoids, flavonoids, folic acid, vitamins and fiber. On the other hand, omega-3 fatty acids contained in plants (α-linolenic acid (ALA)), but also in fish oil and supplements, such as DHA and EPA, constitute blood cell membranes and are essential for normal brain function [111,112]. These compounds may act through multiple pathways. They can reduce lipid peroxidation, protein and DNA oxidation, modulate the immune system and regulate synaptic activity and axonal density [113]. Several studies also suggest that they mediate microglia cells through the gut-brain axis, thus regulating the release of pro-inflammatory cytokines, which have a detrimental role in cognitive impairment [114,115]. DHA also stimulates neuronal plasticity through the Akt pathway and the neurotrophin BDNF, a signaling molecule that modulates synaptic plasticity and protects against neuronal apoptosis [116], but also influences energy metabolism, insulin sensitivity and glucose and lipid metabolism [117]. BDNF has been shown to increase in brain areas related to AD after the consumption of plant foods and supplements, as well as fermented foods [118]. For example, DHA supplementation has been found to elevate levels of hippocampal BDNF and enhance cognitive function in mice [119]. Besides facilitating synaptic plasticity and membrane fluidity, omega-3 fatty acids might also exert an effect on metabolism, stimulating glucose utilization [120], mitochondrial function [121] and reducing ROS [119].

Oxidative stress leads to vascular dysfunction and cell death. Adherence to the Mediterranean diet reduces vascular risk factors, such as LDL cholesterol and increases favorable HDL cholesterol [122], reduces systolic blood perfusion, exerts vasodilation and might promote angiogenesis. Moreover, components of this diet, such as legumes and whole-grain foods, may lower the glycemic index and improve cognitive function, which is of particular interest in the case of diabetic patients to cease the infiltration of inflammatory mediators into the brain [123].

Overall, the components of the Mediterranean diet may have a beneficial effect in ameliorating the risk factors for metabolic syndrome, thus contributing to better cognitive performance. In addition, high-fiber foods may have indirect effects on cognition through regulating intestinal activity [124] and facilitating nutrient absorption, for example that of GABA. Furthermore, the combination of various micronutrients in foods and supplements, such as antioxidants and flavonoids, might have an additive or synergistic effect on cognition. Of special interest are also the changes on brain integrity, as revealed by structural, function and metabolic changes. Brain image could be a surrogate marker of cognitive changes associated with nutrition, but few in this review have used them [27,28,37,41,50,64,65,72]. Since vascular dysfunction and synaptic loss are common features of cognitive impairment, increase in blood perfusion, glucose metabolism and neuronal activation might be good biomarkers of cognitive impairment.

Lastly, there are sex-specific differences in the effect of some nutrient components, such as DHA [125]. These differences could be related to sex hormones, as they influence PUFA metabolism differently.

### 4.8. Limitations and Strengths

There are several limitations that need to be taken into consideration when assessing the findings. Firstly, the dietary interventions were highly heterogeneous, with a wide age range of participants, various geographic regions, different durations of interventions of which many with short follow-up periods, and different measures of cognitive functioning, making it difficult to compare the effectiveness of the interventions across trials. Secondly, the characterization of subjects into cognitively healthy, SCD and cognitively impaired is not always well established in the trials, and even between articles there may be some variation when classifying individuals. It is possible that a fraction of MCI and SCD individuals might by AD patients, but in almost all the studies they used these terms in an unspecific way, without biomarkers of AD to support it and/or measured using a MMSE/MoCa or similar test. Third, it is noteworthy that most of the studies dealt with supplement interventions, many of them of plant origin, as well as that more than half of the studies were conducted in cognitively healthy participants. Moreover, in this systematic review multimodal interventions were excluded because of the complexity of disentangling observed effects. However, several multimodal interventions (i.e., MAPT [126], FINGER [127]) have been relatively successful in combating complex chronic disorders. Overall, as reflected in the RoB and Figure 6, many studies judged at high risk of bias were conducted with a small sample size. Therefore, larger and high-quality studies are needed to avoid potential false positives.

The present systematic review also has strengths. Firstly, it presents the most comprehensive and updated assessment of this topic; secondly, it examines dietary patterns in people who are not diagnosed with dementia. This might be a useful strategy in relation to other nutrients, since it prevents dementia from a manageable approach easy to implement in dietary habits.

### 4.9. Further Study Directions

Future RCTs could test other types of dietary patterns, such as vegetarianism or pesco-vegetarianism, and foods such as dairy products or seafood/fish, among subjects with SCD and cognitive impairment. Additionally, it would be interesting to study gender responses to the interventions, and the potential modulatory role of gut microbiota composition to better establish the gut–brain-axis system [128].

## 5. Conclusions

Oxidative stress and inflammation may play an important role in cognitive function in healthy subjects, those subjects with cognitive impairment and self-perceived cognitive decline. Dietary counselling interventions, food-based interventions and dietary supplements with antioxidant and anti-inflammatory properties may help to ameliorate inflammation and oxidative damage.

The studies included in this review support that healthy dietary patterns, specific foods and dietary supplements can (a) improve memory, language, attention and concentration, executive functions, psychomotor speed and further cognitive domains, and (b) increase blood perfusion in areas typically related to AD.

Taken together, the existing evidence from RCTs over the last four years is in line with most of the previous findings already reported in the preceding systematic reviews. It confirms the beneficial role of the Mediterranean diet, plant foods and protein-rich supplements, some amino acids and minerals, polyphenols and the combination of them, as well as other types of dietary supplements. In addition, it supports the need for further research, especially in terms of vitamins and PUFAs, since different effects on cognitive function have been reported, but also other types of nutritional supplements which have been little studied. Moreover, most of the studies evaluated cognition, but it would also be necessary to determine brain changes, which could be useful to identify the mechanisms behind these benefits.

These data highlight the need for nutritional strategies that are most appropriate in maintaining cognitive function and protecting against age-related and AD cognitive impairment, thus reducing the risk, delaying the onset or slowing the progression of cognitive impairment and dementia in healthy, SCD and cognitively impaired subjects.

## Figures and Tables

**Figure 1 nutrients-13-03728-f001:**
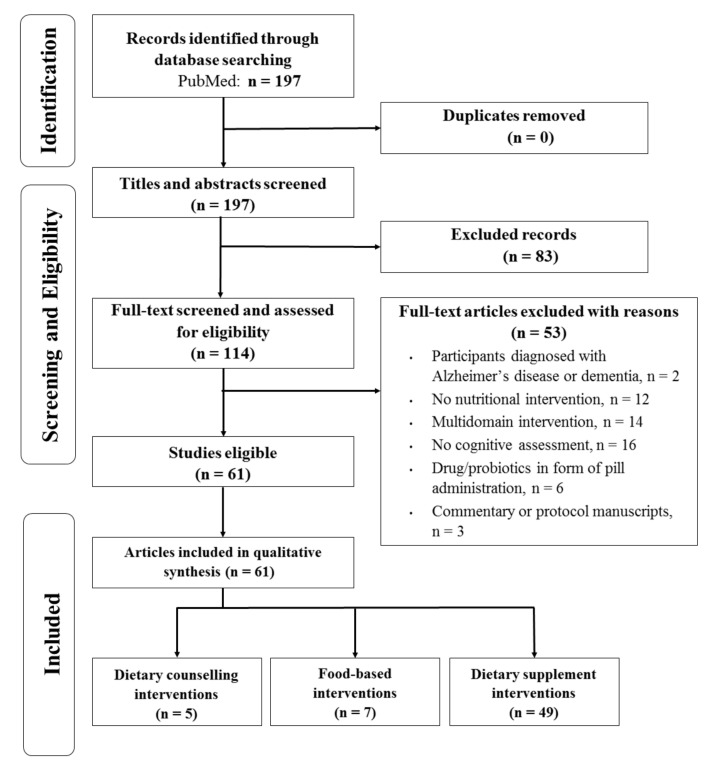
Flow diagram of studies assessed for eligibility per screening stage.

**Figure 2 nutrients-13-03728-f002:**
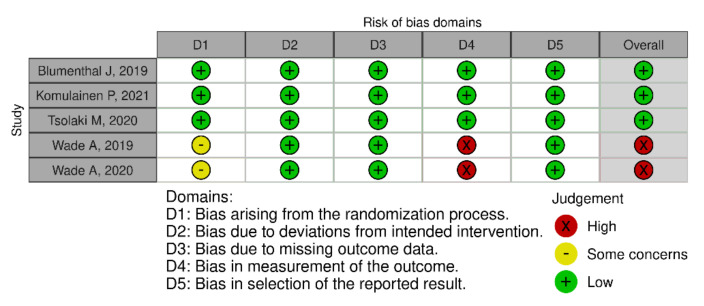
Quality assessment of the RCTs that followed a diet counselling intervention.

**Figure 3 nutrients-13-03728-f003:**
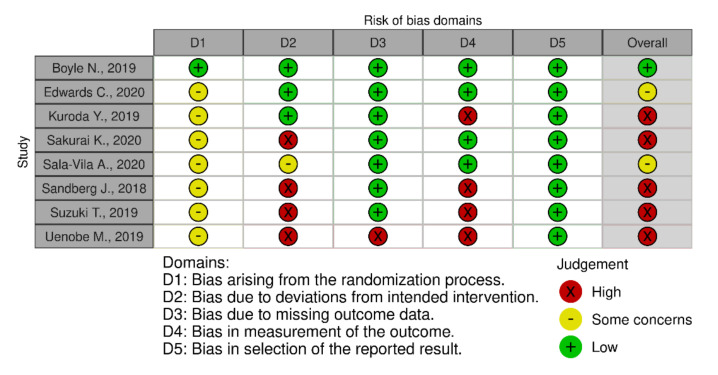
Quality assessment of the RCTs that followed a food-based intervention.

**Figure 4 nutrients-13-03728-f004:**
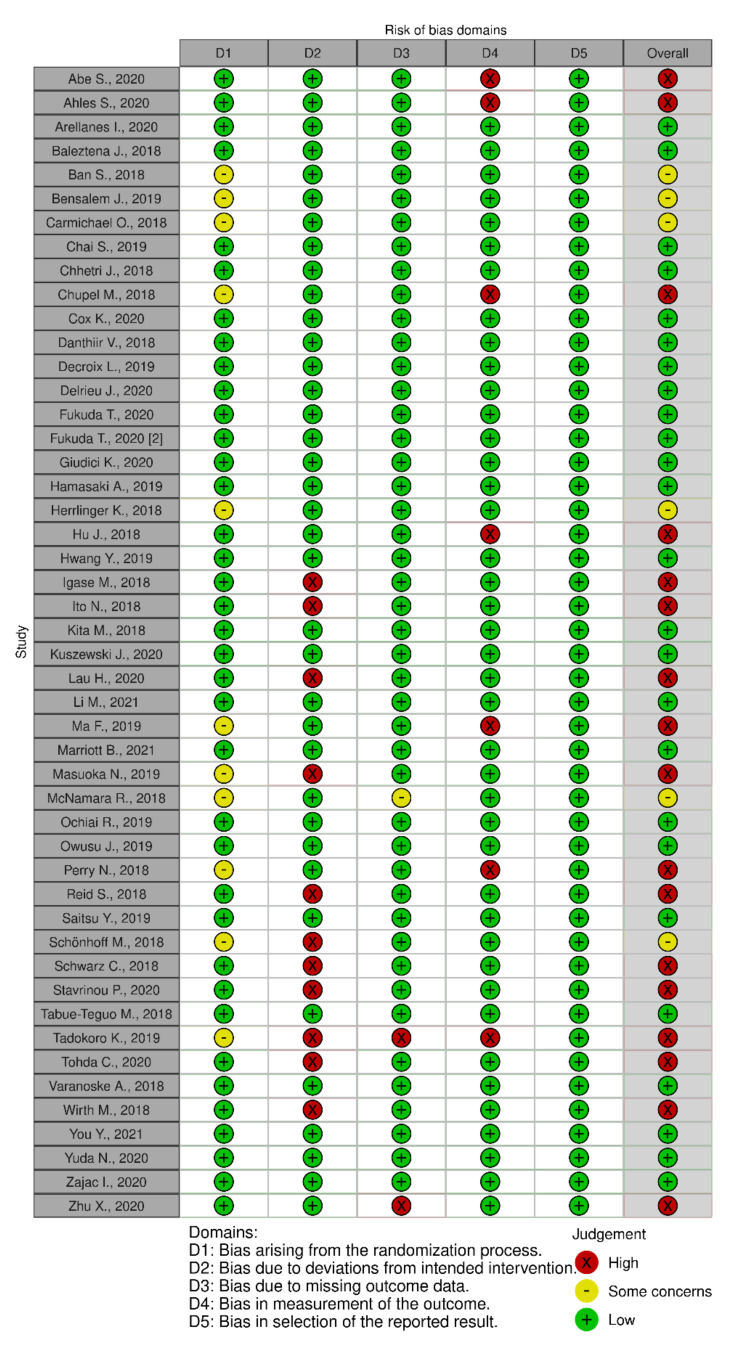
Quality assessment of the RCTs that followed a dietary supplement intervention.

**Figure 5 nutrients-13-03728-f005:**
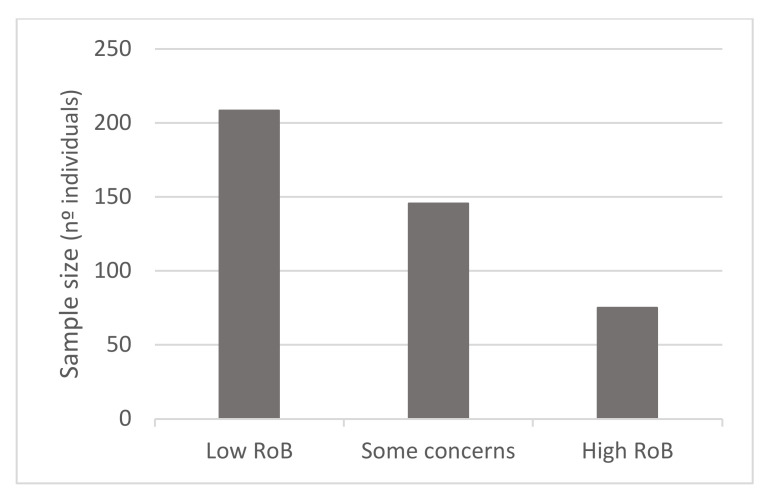
Mean values of the studies’ sample size according to the RoB Sample size range: 19 to 843 in low RoB, 11 to 708 in some concerns, 21 to 250 in high RoB.

**Figure 6 nutrients-13-03728-f006:**
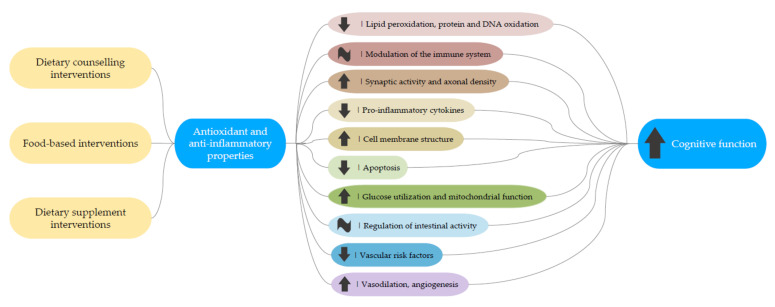
Potential mechanisms linking dietary interventions with cognitive function improvements.

## Data Availability

No new data were created or analyzed in this study. Data sharing is not applicable to this article.

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
