# Peer review of "Effects of Nutrition on Cognitive Function in Adults with or without Cognitive Impairment: A Systematic Review of Randomized Controlled Clinical Trials"

_nutrients, 2021, doi:10.3390/nu13113728_

Round 1

Reviewer 1 Report

The topic of the review is interesting and read well. My only concern is that US department of agriculture has done a system review of the same.

Reviewer 2 Report

In the present Systematic Review, the Authors conducted a analyzed of the randomized controlled trials published in MEDLINE PubMed from January 2018 to July 2021 which investigating the impact of dietary counselling, food-based and dietary supplement interventions on cognitive function in adults with or without cognitive impairment. In my opinion, the manuscript is very well prepared, with clearly described methods of including research into the analysis.

The manuscript is very interesting and informative. However, there is one point of concern, which should be properly addressed to further improve the quality of the manuscript. 

I propose the following changes:

  • In the discussion subsection "4.7 Potential Mechanism" I propose to prepare a scheme of the mechanism of action of the discussed compounds. 
  • In the Summary chapter, I also propose a summary diagram in which the authors will mark all things known and requiring further research.
